# Relationship between the Rate of Perceived Stability, electrodermal activity and task performance during balance challenges in chronic stroke

Aishwarya Shenoy[1,2,3], Amy Schneeberg[4], Towela Tembo[1,3], Rebecca M. Todd[5,6], Noah D. Silverberg[1,5], Janice J. Eng[1,2,3], Tzu-Hsuan Peng[1,3], Peyman Servati[7], Courtney L. Pollock[1,2,3]*

1 Rehabilitation Research Program, Centre for Aging SMART and GF Strong Rehabilitation Centre, Vancouver, British Columbia, Canada, 2 Graduate Program in Rehabilitation Sciences, University of British Columbia, Vancouver, British Columbia, Canada, 3 Department of Physical Therapy, University of British Columbia, Vancouver, British Columbia, Canada, 4 Department of Occupational Science and Occupational Therapy, University of British Columbia, Vancouver, British Columbia, Canada, 5 Department of Psychology, University of British Columbia, Vancouver, British Columbia, Canada, 6 Djavad Mowafaghian Centre for Brain Health, Vancouver, British Columbia, Canada, 7 Department of Electrical and Computer Engineering, University of British Columbia, Vancouver, British Columbia, Canada

* courtney.pollock@ubc.ca

## Abstract

### Introduction

In addition to sensorimotor impairments following stroke, decreased self-efficacy regarding walking balance may lead to self-imposed limitations on community level mobility, especially among women. The Rate of Perceived Stability (RPS) is a self-efficacy measure used to assess individual perception of balance ability when standing or walking balance is challenged. Measurement of electrodermal activation (EDA), modulated by the autonomic nervous system, during perturbations to standing balance reflects the physiological arousal ('fight or flight') response of the individual as they maintain or recover their balance. Repeat performance of a balance task has been shown to result in habituation of EDA within a single session; however, studies have yet to test whether similar habituation occurs when the same balance tasks are repeated across different days. This study aims to examine the relationships between EDA, task performance ability, and RPS in individuals with chronic stroke performing walking balance challenges. Further, the study explores how sex and repeat expo-sure (repeat performance of task) moderate these relationships.

### Methods

Over two testing days, participants with chronic stroke (>1 year) were assessed on walking balance task performance with the Community Balance and Mobility Scale

**Data availability statement:** The study data used for data analysis are publicly available in the Borealis Dataverse Repository at the following DOI: https://doi.org/10.5683/SP3/SXVBGQ.

**Funding:** This work was supported in part by grants from the New Frontiers in Research Fund (Exploration grant, CLP) and the Michael Smith Foundation for Health Research (Scholar Award, CLP). The funders had no role in study design, data collection and analysis, decision to publish, or preparation of the manuscript.

**Competing interests:** The authors have declared that no competing interests exist.

(CB&M) and rated their perceived stability using the RPS. EDA measured the physiological arousal during task performance. Linear mixed models were used to assess: 1) the relationship between CB&M task performance and RPS and whether sex or repeat exposure moderates this relationship, 2) the relationship between the physiological arousal response and RPS and whether sex or repeat exposure moderates this relationship, and 3) whether physiological arousal response mediates the relationship between CB&M task performance and RPS.

## Results

Thirty individuals with chronic stroke, with moderate severity lower extremity impairment (Chedoke McMaster Stroke Assessment score 4–5/7) participated in the study, including 15 males (mean age: $65.1 \pm 10.2$ years; time since stroke: $9.4 \pm 4.7$ years) and 15 females (mean age: $65.5 \pm 9.7$ years; time since stroke: $7.6 \pm 5.9$ years). CB&M scores, indicating balance performance, explained 20.3% of the variability in the RPS. As CB&M task performance improved, RPS scores decreased by 2.69 (95% CI [−3.28 – −2.10]) to 3.67 (95% CI [−4.32 – −3.02]) points, indicating improved perceived stability. Physiological arousal significantly predicted RPS scores, however only explained 1.6% of the variability in the RPS. Physiological arousal was not found to be a significant mediator of the relationship between the CB&M task performance scores and RPS. Participant-specific random effects accounted for more variance in the RPS than the fixed effects of task performance and physiological arousal, explaining 46% of variance in RPS. Repeat exposure and sex did not moderate the relationships between the predictors (physiological arousal and task performance) and RPS.

### Conclusion

Ability to perform a walking balance task (CB&M task performance score) and the underlying physiological arousal response (EDA) are independent predictors of perception of balance in people with chronic stroke as measured by the RPS. However, individual characteristics not captured in this study account for a greater proportion of the variability of the self-reported perception of balance during tasks performed. Potential characteristics may include constructs such as fall history and level of physical activity highlighting the complexity of perception of balance ability post-stroke.

### Introduction

Walking ability is a key factor in achieving independence and mobility associated with community re-integration post stroke [1]. Retraining walking balance is a priority during stroke rehabilitation with independent walking achieved by ~80% of people post stroke [2]. After discharge from inpatient stroke rehabilitation, balance confidence predicts individuals' perceived physical function, mobility, and further recovery

[3–5]. Indeed falls have been reported to occur in 50% of community dwelling stroke survivors [6]. Optimizing rehabilitation of walking balance beyond basic walking function remains a critical need.

Rehabilitation of walking balance requires balance to be challenged near the limits of the individual's capacity to induce a training effect [7]. Recently, Espy et al [8] developed the Rate of Perceived Stability (RPS) (Fig 1) to address the lack of a measure for balance exercise intensity and aid clinicians in prescribing and progressing the difficulty of balance exercises. The ratings on the RPS reflect the individual's perception of their ability to maintain their balance and prevent a fall. We recently established the RPS as a valid and reliable measure of challenge to walking balance in ambulatory people with stroke [9]. However, various factors can impact patient-reported ratings, such as social desirability bias [10] and patient perception of their balance abilities which may be informed by their self-efficacy and history of falls [4,11]. Understanding the relationship between self-reported perception of balance using the RPS and clinical measurement of task performance while performing varied levels of walking balance challenges may further our interpretation and use of RPS scores to inform rehabilitation of walking balance.

Heart rate variability (HRV), cortisol production and electrodermal activity (EDA) can measure physiological arousal modulated by the autonomic nervous system (i.e., "fight or flight" response) [12,13]. EDA is a measurement of skin conductance driven exclusively by sympathetic nervous system activation [14], and is activated in conditions requiring heightened attention, state anxiety, or cognitive load [15]. EDA has shown to be a better metric to capture brief, task-related changes in sympathetic nervous system activity than metrics specific to HRV and cortisol production [16]. Previous studies have reported increased EDA in people with stroke when walking and standing balance is challenged, such as during postural perturbations and walking while navigating obstacles [17,18]. EDA has also shown to correlate with "subjective units of distress (SUDS)", a common metric in clinical psychology similar to the RPS in that it captures subjective experience of distress during exposure-based therapies [19,20]. It is possible that EDA may hold a similar relationship with RPS which represents subjective experience of balance challenge. Additionally, decline in task performance in people with chronic stroke has shown to result in increased physiological arousal response [17] and increased RPS ratings [9]. Taken together, this suggests potential relationships between level of task performance, the physiological arousal response and reported RPS scores during tasks that challenge standing and walking balance in people with stroke. However, no studies have explored these relationships which may provide further insights regarding the clinical utility of the RPS scale. Furthermore, as walking balance interventions extend over multiple sessions, it would be important to determine whether these relationships change with repeated exposure to the walking challenges on subsequent days to enhance the clinical utility and interpretability of treatment intensity measures. Repeat performance of a balance task has been shown to result in habituation of EDA within a single session [18]; however, studies have yet to test whether similar habituation occurs when the same balance tasks are repeated across different days.

| Steady | Balance does not feel challenged, but may have some body movements | 2 |
| | | 3 |
| Unsteady | Feels like work to keep balance, but still do not need to step OR reach | 4 |
| | | 5 |
| Mildly Unbalanced | Feels like I might take a step OR reach for support to maintian balance | 6 |
| Moderately Unbalanced | | 7 |
| Unbalanced | Feel that even the smallest or sudden movement will cause a fall | 8 |
| Very Unbalanced | | 9 |
| About to Fall | Extremely challenged; have to step AND/OR grab support to keep balance | 10 |

**Fig 1. The Rate of Perceived Stability (RPS) Scale is a 10-point self-rating scale designed to provide clinically meaningful descriptions of stability.** It allows individuals to assess their perceived ability to maintain balance during a given task.

Sex differences have been previously reported in stroke subtypes, severity and recovery outcomes [21,22]. Although males have higher incidence of first-ever stroke compared to females [23], prognosis and recovery outcomes related to walking are reported to be poorer in females than males despite adjustment for baseline differences in age, pre-stroke function and comorbidities [22,24,25]. Previously, we showed that while females with stroke reported lower balance confidence than males, they interpreted and utilized the RPS similarly during balance-challenging tasks or task performance [9]. Prior studies monitoring changes in the physiological arousal response during balance challenges in people with stroke have not explored potential sex differences [17,18,26,27].

The overarching aim of this study is better understand the construct measured by the RPS, perceived stability, in individuals with chronic stroke. This aim is addressed through three sequential objectives: 1) investigate the relationship between walking balance task performance scores and RPS ratings, and determine whether sex or repeated exposure (repeat performance of tasks) across days moderates this relationship, 2) examine the relationship between the physiological arousal response (EDA) and RPS ratings during walking balance tasks and assess whether sex or repeated exposure across days act as moderators of this relationship; and 3) explore potential mediation of the relationship between walking balance task performance and RPS by the physiological arousal response. We hypothesize that, 1) walking balance task performance scores (as measured by items from the Community Balance and Mobility Scale (CB&M)) will predict RPS scores, and that sex or repeated exposure across days will not moderate this relationship; 2) the physiological arousal response will predict RPS scores, with sex or repeated exposure acting as moderators of this relationship; and 3) the relationship between walking balance task performance and RPS scores will be partially mediated by the physiological arousal response, such that lower physiological arousal in those with better task performance accounts for lower RPS.

## Materials and methods

### Experimental protocol

Thirty people with stroke (>1 year) were recruited from the local community. Participant recruitment began on July 10,2019 and ended on February 10, 2020. Inclusion criteria included: at least one year-since stroke (occurrence of stroke confirmed by medical records), hemiparesis post-stroke, the ability to walk at least 10 metres with or without a walking aid (confirmed during initial assessment). Participants with expressive aphasia were included with adaptations to verbal reporting of RPS scores as needed. Individuals were excluded if, in addition to stroke, they had any health conditions that could limit their involvement in regular walking activity (e.g., neurological conditions unrelated to stroke, severe arthritis, cardiac or respiratory conditions). Participants provided informed written consent before participation. This study was approved by the University of British Columbia Clinical Research Ethics Board.

### Participant and clinical characteristics

Descriptive measures of participants included age, time since stroke, type of stroke, severity of lower extremity impairment (Chedoke McMaster Stroke Assessment (CMSA)) [28]. Anxiety was measured using the State Trait Anxiety Inventory (STAI) [29] and balance confidence was measured using the Activities-specific Balance Confidence scale (ABC) [30]. The CMSA is scored on a 7-point scale and is administered by a physiotherapist. Stage 0 is summarized as the presence of flaccid paralysis and a score of 7 is summarized as normal movement [28].The ABC is a self-report questionnaire in which individuals rate their degree of confidence in their ability to perform common activities within their house and community on a scale of 0% (no confidence) to 100% (extremely confident) [30], where scores between 50–80 indicate medium level of functioning and higher than 80 indicate high level of functioning [31]. Both scales have shown to be reliable and valid in people with stroke [28,32]. The state (temporary anxiety) and trait (long-term anxiety) sections of the STAI are each out of 80, a score between 20–37 indicates no to low anxiety, 38–44 moderate anxiety and 45–80 as high anxiety [29].

 

### Challenging walking and standing balance

Participants attended two data collection sessions, with the second occurring within 2–10 days of the first. Both days followed the same experimental protocol, except for descriptive clinical measures (ABC, CMSA – Leg and Foot, STAI) that were only collected on Day 1.

On Day 1 and Day 2, walking and standing balance were challenged by asking participants to perform tasks from the CB&M scale. The CB&M is a valid, reliable measure of post-stroke walking balance, with tasks ranging in difficulty making it sensitive to changes in functional balance and mobility without showing a ceiling effect [33,34]. Participants completed all but two tasks, "Walk, Look and Carry" (walking while looking to the side and carrying weighted bags in both hands) and "Descending Stairs" due to equipment requirements and varying levels of upper extremity motor function in our participants. Participants were scored on their performance for each task by a physiotherapist using the CB&M scoring guidelines (0 unable; 5 unimpaired performance). During data collection, some participants either declined to attempt tasks they perceived they would be unable to perform or performed tasks in a manner that did not adequately challenge their balance (i.e., use of physical assistance), leading to a CB&M score of 0. CB&M task performance scores originally on a scale of 0–5, were regrouped to 4 categories. We collapsed across scoring in-line with findings of a Rasch analysis of the CB&M Scale used in people with stroke reporting disordered thresholds of the 0–5-point scoring scale which was resolved by collapsing scores [34]. The recategorized categories are as follows:

- 0 for complete inability (Category 1)

- 1–2 for poor performance (Category 2),

- 3 for moderate performance (Category 3)

- 4–5 for high performance (Category 4).

CB&M items were only included in analyses if 1) at least 10 participants completed the task (to ensure enough observations were present) and 2) the mean CB&M score was at least 2 (prior to recategorization), indicating that on average, participants performed the task without physical assistance.

### Subjectively measuring perception of balance challenge

Upon completion of each task participants were asked to rate their perceived stability using the RPS scale (Fig 1).

### Physiological Arousal Response (EDA)

Prior to applying the electrodes, the skin was cleaned using water. Two electrodes were applied on the palmar surface of the non-paretic hand as the physiological arousal response has shown to be supressed on the paretic side [35]. The physiological arousal response was collected as a measure of skin conductance with a current of 50 mV applied between the two electrodes at a sampling frequency of 1024 Hz (Cambridge Electronics Design Ltd., UK). Before commencing each task, a one-minute quiet-stance baseline of physiological arousal response was collected. Event markers were used to capture the start and the end of each task.

### Signal processing and analysis

Physiological arousal response signals were processed and analyzed using Spike 2.0 (Version 6). A 4-point moving average of the raw signal was calculated to remove signal noise. For each task, the pre-task baseline was calculated as the four second mean activity that occurred around the minimum value (2 seconds before and 2 seconds after the minimum) during the pre-task quiet stance [18]. Manual inspection was employed to ensure the epoch representative of the pre-task baseline was stable and did not include large fluctuations in the signal. The task-evoked physiological arousal response

for each task was calculated as the five second mean activity that occurred at the peak physiological arousal response (2.5 seconds before the peak and 2.5 seconds after the peak) during task performance (Fig 2). The task-specific physiological arousal response for each item in the CB&M was then calculated by subtracting the pre-task baseline from the corresponding task-evoked physiological arousal response. While larger EDA responses reflect greater autonomic arousal within individuals, it is difficult to compare the responses between subjects as baseline levels and response amplitudes can vary between people due to non-task related factors (e.g., medications, skin properties) [36]. Therefore, for each day, the maximum task-specific physiological arousal response out of all the tasks was identified for each participant separately. This participant-specific maximum response was used to normalize response values across all tasks performed by the individual to aid interpretation. Task-specific physiological arousal response values after normalization ranged from 0%−100%, with 100% representing the task that triggered the participant's maximum response for that day. The normalized task-specific physiological arousal response for each item in the CB&M was used for data analyses. This preserved within-person response patterns while enabling group-level modeling of the relationship between EDA and perceived stability (RPS) across tasks.

## Statistical analyses

Statistical analysis was conducted using R Studio software (Version 4.1.1) [37]. Descriptive statistics were conducted for all variables measured. A MANOVA was used to examine the effect of sex on the clinical characteristics amongst participants.

**Linear Mixed Models (LMM).** To address Objectives 1 and 2, Linear Mixed Models (LMMs) using the lmer function from the lme4 package [38]. Two different models were used for each objective, where the predictor (independent

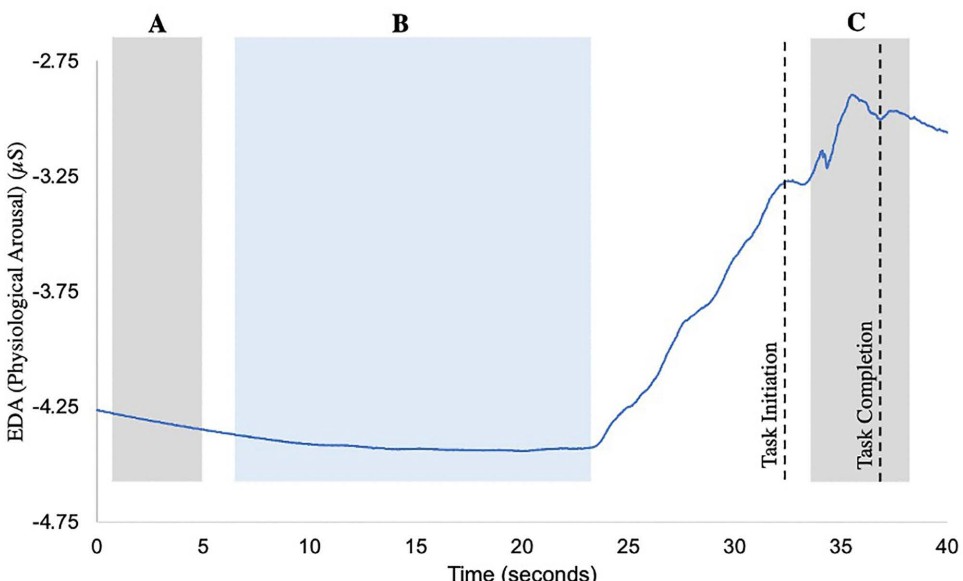

**Fig 2. Physiological arousal response signal (blue line) collected performance of balance task of the Community Balance and Mobility Scale (180° Tandem Pivot task). A.** Pre-task baseline activity was calculated as the 4-second mean centered around the lowest stable point during the standing baseline period (shaded grey). **B.** Represents the interval when the participant received task instructions (shaded blue). Black dashed lines mark task initiation and completion. Task initiation was defined as the onset of physical movement, while task completion was marked by the end of movement following task execution. **C.** The response to task performance was analyzed within a 5-second window spanning 2.5 seconds before and after the peak physiological arousal following initiation of task execution (shaded grey). The peak response occurred after task completion, resulting in the window extending beyond task completion to account for known potential delays of up to 4 seconds in measurement of physiological arousal response.

variable) was CB&M Task Performance for Objective 1 and physiological arousal for Objective 2. For both models (Model 1. CB&M task performance and Model 2. Physiological arousal for each CB&M task), each CB&M task was treated as a separate observation. Because multiple tasks were completed by each participant, the data had a hierarchical structure with tasks nested within participants. This grouping structure was modeled by including a random intercept for participant to account for within-subject correlation. Sex and repeated exposure (Day 1 vs Day 2) were explored as potential effect moderators by including them as separate interaction terms in both models, nested models were compared via likelihood ratio tests ($p > 0.05$ considered significant). Tukey post hoc test was performed to assess pairwise differences between groups for all significant interaction terms. Post hoc pairwise comparisons among CBM performance levels were conducted in R using the emmeans package with Tukey's HSD adjustment, which corrects for multiple comparisons by controlling the family-wise error rate at $\alpha = 0.05$. Likelihood ratio tests were used to compare models with and without the interaction terms, which were retained if the p-value was $< 0.10$.

**Mediation analysis.** To address Objective 3, a mediation analysis was conducted to investigate whether CB&M task performance influenced RPS scores were significantly accounted for by the physiological arousal response (Fig 3). The analysis was carried out using the `mediation` package in R, which directly estimates the indirect effect of the independent variable (CB&M task performance) on the dependent variable (RPS scores) through the mediator (physiological arousal response). Power for the mediation pathway was approximated by calculating power for each constituent effect (CB&M task performance→ physiological arousal; physiological arousal→RPS) using Cohen's $d$ and the pwr package, then combining them. Observed, standardized, and hypothetical medium-sized effects ($d = 0.35$) were evaluated for sensitivity. The indirect effect was estimated using bootstrapping with 5000 resamples. The significance of the indirect effect was assessed by checking whether the 95% bias-corrected confidence interval for the indirect effect excluded zero [39]. The direct effect of CB&M task performance on RPS scores was also assessed.

## Results

### Participants

Thirty participants (15 males and 15 females) with chronic stroke consented to participate (Table 1). Four participants (2 male) had expressive aphasia and used both strategies of verbal reporting and pointing to report RPS scores. Two participants were unable to complete tasks on Day 1 due to lower levels of functional abilities; therefore they did not return for Day 2 and one participant's data showed poor signal quality. These 3 participants were excluded from analysis (n = 27) (Fig 4).

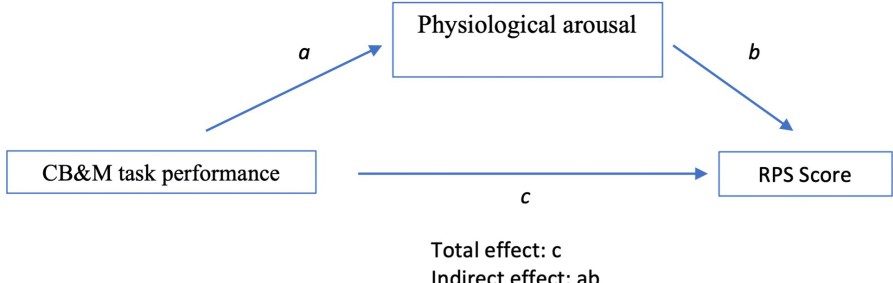

**Fig 3. Mediation analysis showing the role of the physiological arousal response as a mediator in the relationship between CB&M task performance scores and RPS scores.** Path (*a*) represents the effect of CB&M task performance on the physiological arousal response, while path (*b*) represents the effect of the physiological arousal response on RPS scores. The product of these paths (ab) indicates the indirect effect, capturing the extent to which physiological arousal response mediates the relationship between task performance and RPS. Path (*c*) represents the total effect of CB&M task performance on RPS scores, encompassing both direct and indirect effects.

**Table 1. Participant characteristics.**

| Participant characteristics | Females | Males |
|---|---|---|
| Age (y, mean (SD) | 65.5 (9.7) | 65.1 (10.2) |
| Time since stroke (years, mean (SD) | 7.6 (5.9) | 9.4 (4.7) |
| Type of stroke (Ischaemic/haemorrhagic/unknown) | 9/4/1 | 10/4 |
| Hemiparetic side (right/left) | 7R/7L | 6R/8L |
| Use of Walking Aid (none/cane or pole/walker) | 8/4/2 | 5/9/0 |
| Ankle-foot orthosis (none/fixed/flexible) | 9/1/4 | 10/2/2 |
| *Clinical Measures*[a] | | |
| Comfortable Gait Speed (m/s) | 0.7(0.3) | 0.8 (0.3) |
| Six Minute Walk Distance (m) | 283.24(109.6) | 329.33(129.4) |
| ABC (/100) | 66.3 (20.8) | 81.9 (13.6) |
| CMSA (foot/leg)[b] | 4.5 (2)/5.8 (1.3) | 4.3 (2)/5.2 (1.6) |
| CB&M total score (/80) | 26.8 (12.8) | 25.7 (16.5) |
| STAI State (/80) | 34.46 (10.44) | 28.57 (7.94) |
| STAI Trait (/80) | 35.08 (9.20) | 34.36 (9.36) |

[a]All clinical measures results are presented as mean (SD).

[b]Chedoke McMaster Stroke Assessment (CMSA), scores for one participant not collected.

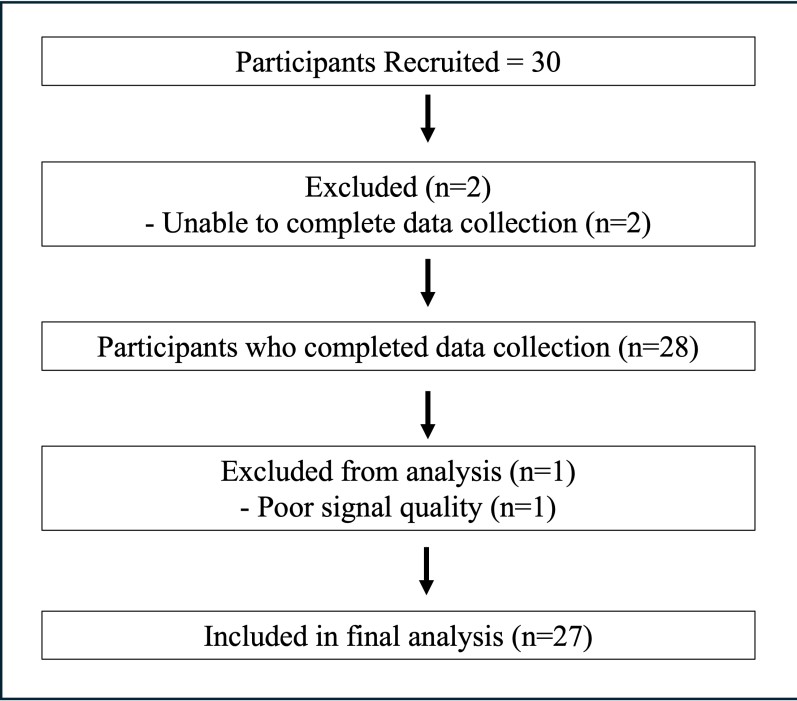

**Fig 4. Flowchart illustrating participant recruitment, reasons for exclusion, and attrition throughout the study, including the final number of participants analyzed.**

The MANOVA showed that there was no significant difference between males and females in age ($p = 0.82$), years since stroke ($p = 0.26$), level of motor impairment as measured by the CMSA-Foot ($p = 0.90$) and CMSA-Leg ($p = 0.37$), and level of anxiety as measured by STAI-state ($p = 0.11$) and STAI-trait ($p = 0.84$) (Table 1). The only sex difference was on the ABC, with females scoring lower balance confidence (66.3%) than men (81.9%; $p = 0.02$).

Mean CMSA-Foot and CMSA-Leg ranged between 4–5/7, which represents a moderate level of impairment of the foot and leg after stroke. No difference in STAI-state ($p = 0.72$) anxiety was seen between days and reflected no to low anxiety amongst participants at the initiation of data collection on each separate day.

## CB&M Tasks

A total of 11 of 16 CB&M tasks were included (Table 2). Three tasks, in which fewer than 10 participants attempted to perform the task, were excluded (Lateral Foot Scooting on the paretic leg, Hopping Forward on the paretic leg and Running with Controlled Stop). Two tasks with an average CB&M performance score less than 2 were also excluded (Unilateral Stance on the paretic leg and Hopping Forward on the non-paretic leg). Removing these tasks from analysis ensured that the corresponding physiological arousal response to a task was specifically attributable to balance being challenged. A total of 583 observations of CB&M task performance scores from 27 participants were included. There was no difference between females and males in total performance scores on the CB&M scale ($p = 0.85$; Table 1).

## Linear Mixed Model (LMM)

**Objective 1: Task performance as a predictor of RPS and Day as a moderator.** Higher levels of CB&M task performance scores were associated with lower RPS scores. Compared to the CB&M score of 1, CB&M scores of 2, 3, and 4 were associated with a decrease of 2.80 (95% CI: −3.29 to −2.30, $p = 0.001$), 2.69 (95% CI: −3.28 to −2.10, $p = 0.001$), and 3.67 (95% CI: −4.32 to −3.02, $p = 0.001$) points in the RPS ratings, respectively.

Post hoc pairwise comparisons revealed that the difference in RPS scores between CB&M task performance = 1 and CB&M = 2, and CB&M = 3 and CB&M = 4 were both statistically significant ($p = 0.001$), whereas the difference between CB&M = 2 and CB&M = 3 was not statistically significant ($p = 0.95$; see Fig 5).

Model Fit: The marginal $R^2$ for the model was 0.203, indicating that the fixed effect of CB&M task performance scores explained 20.3% of the variance in RPS. The conditional $R^2$ was 0.451, suggesting that participant variability (the random effect) accounted for a larger portion (45.1%) of the variance in RPS.

Effect Moderators: The likelihood ratio tests revealed that including interaction terms between task performance and sex (Model 2B; $\chi^2 = 1.63$, $p = 0.80$) and day (Model 2C; $\chi^2 = 6.03$, $p = 0.20$) did not significantly improve model fit. These

**Table 2. Tasks included from the Community Balance and Mobility (CB&M) Scale.**

| CB&M Tasks |
| --- |
| 1. Unilateral Stance Non-Paretic |
| 2. Tandem Walking |
| 3. 180 Tandem Pivot |
| 4. Lateral Foot Scooting Non-Paretic |
| 5. Crouch and Walk (pick up objects from the floor) |
| 6. Lateral Dodging (crossover steps) |
| 7. Walking and Looking Right |
| 8. Walking and Looking Left |
| 9. Forward to Backward Walking |
| 10. Step Ups Non-Paretic |
| 11. Step Ups Paretic |

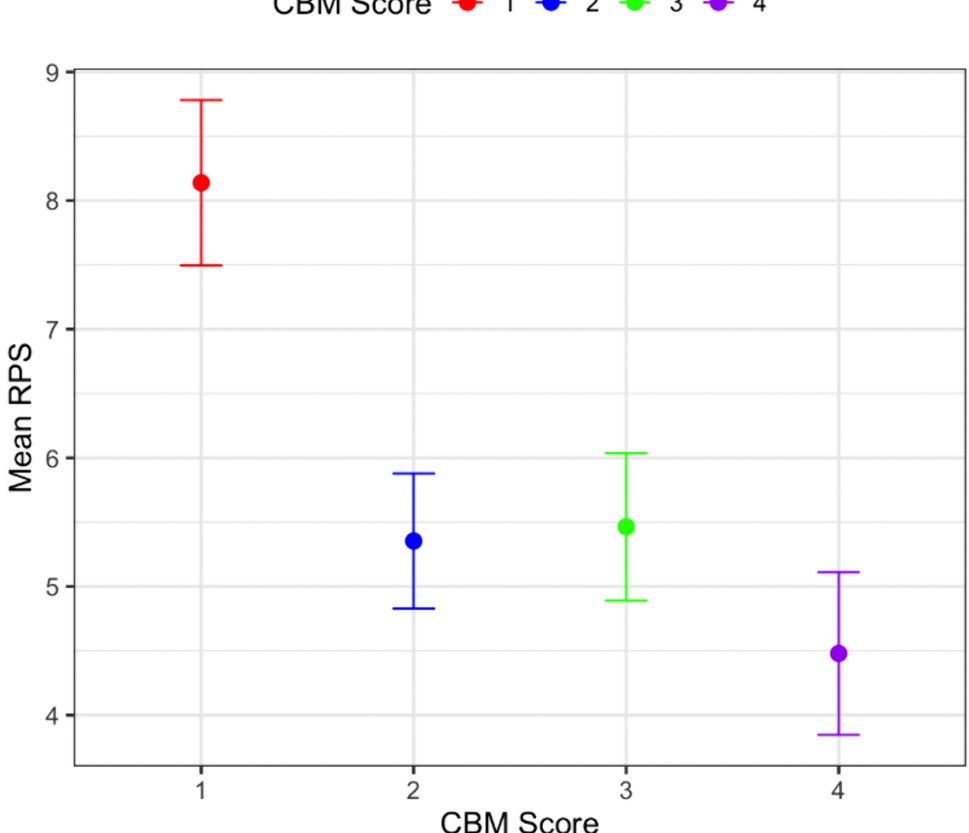

**Fig 5. Estimates of mean RPS from LMM model with CB&M scores.** * CB&M task performance scores of 2,3, and 4 are statistically significantly different (p < 0.1) compared to CB&M Score 1.

findings suggest that the relationship between CB&M task performance and RPS is consistent across sex and does not vary with repeated exposure over the two testing days.

**Objective 2: Physiological arousal response as a predictor of RPS and Day as a moderator.** Physiological arousal response was positively associated with RPS, ($\beta = 0.13$), indicating that for every 10% increase in physiological arousal response the RPS rating increased by 0.13 units (95% CI [0.06, 0.20], $p = 0.001$).

Model Fit: The marginal $R^2$ for the model was 0.016, suggesting that physiological arousal response explains a small but statistically significant portion (1.6%) of the variance in RPS. The conditional $R^2$ was 0.386, indicating that the random effect (participant variability) accounted for a much larger proportion (38.6%) of the variance in RPS than the fixed effect of physiological arousal response.

Effect Moderators: In examining potential effect moderators, likelihood ratio tests indicated that including interaction terms for sex ($\chi2 = 1.23$, $p = 0.54$) and day ($\chi2 = 1.93$, $p = 0.17$) did not significantly improve model fit. This suggests that the relationship between physiological arousal response and RPS is consistent across sex and does not vary with repeated exposure across the two testing days.

**Objective 3: Mediation of task performance and RPS relationship by physiological arousal response.** The 95% percentile confidence interval for the indirect effect ranged from −0.089 to 0.0546. Since this confidence interval

includes zero, the indirect effect of CB&M task performance on RPS through physiological arousal response was not statistically significant. This suggests that the arousal response does not account for the relationship between CB&M task performance and RPS in this sample.

In a model with both task performance and physiological arousal, both remain significant predictors of RPS. For every 10% increase in physiological arousal response, the RPS rating increased by 0.12 units (95% CI [0.05, 0.18], $p = 0.001$). For task performance, compared to a CB&M score of 1, CB&M scores of 2, 3, and 4 were associated with decreases of 2.78 (95% CI [−3.27, −2.29], $p = 0.001$), 2.78 (95% CI [−3.37, −2.19], $p = 0.001$), and 3.69 (95% CI [−4.33, −3.05], $p = 0.001$) points in RPS ratings, respectively.

Model Fit: The marginal $R^2$ was 0.22, indicating that the fixed effects of physiological arousal response and CB&M task performance together explained 22% of the variance in RPS. The conditional $R^2$ was 0.46, suggesting that, when incorporating both predictors and the random effect (participant variability), they collectively explained 46% of the variance in RPS.

## Discussion

This study aimed to understand the construct measured by the RPS, perceived stability, in individuals with chronic stroke. Lower CB&M task performance scores and higher physiological arousal response were associated with higher RPS ratings. Repeat exposure and sex did not moderate these relationships. However, CB&M task performance accounted for greater variability within RPS scores than the physiological arousal response, and arousal did not account for the relationship between the CB&M task and RPS ratings. These findings suggest that ability to perform a walking balance task as indicated by performance is a stronger predictor of the self-reported RPS than the physiological arousal response. Importantly, participant specific random effects accounted for more of the variability in the RPS scores than either primary independent variable. This highlights the complexity of factors which influence perception of balance function in individuals with chronic stroke.

Despite physiological arousal response being a significant predictor of the RPS, the association was weak. Individual characteristics, emotional states, and changes in alertness or attention influenced by familiarity with the researchers, experimental tasks and environment can also influence the physiological arousal response [14,40,41]. For this reason, our model explored physiological arousal response prediction of RPS scores across days performing the same tasks in the same research environment. Repeat exposure to the testing protocol between days did not significantly change the resulting parameter estimates of the linear mixed model.

Contrary to our hypothesis physiological arousal, as measured by EDA, was not a mediator of the relationship between CB&M and RPS. Task performance was a stronger predictor than the physiological arousal response, suggesting that clinical measurement of walking balance performance, as measured on the CB&M scale, predicts the participants perception of balance ability, as measured by the participant reported RPS scale, irrespective of physiological arousal response during task execution. However, inclusion of participant-specific random effects increased the variability explained in RPS compared to the comprehensive model inclusive of all independent predictors (physiological arousal response and CB&M task performance). This suggests that there are important between individual factors that contribute to the rating of perceived stability during performance of varied walking balance tasks in people with stroke. These individual factors may be shaped by personal experiences with balance impairments and falls which are known to influence balance self-efficacy in people with stroke [4,5]. However, power for the mediation analysis was approximated by calculating it separately for each constituent path and then combining them. This approach, while practical, may overestimate the true power for detecting the mediation effect. Consequently, the mediation analysis should be considered exploratory.

Our findings raise interesting considerations of the utility of physiological stress measures compared to self-reported scalar measures like RPS in the context of retraining walking balance post-stroke. In exposure-based therapy, the relationship between physiological arousal response and SUDS, a participant reported measure of perception of distress,

was studied in two different populations: children with anxiety and adults with social anxiety disorder and eating disorders [19,20]. The findings revealed variability in the association between the physiological arousal response and self-reported SUDS across these studies [19,20] suggesting that individual reported perceptions of anxiety and distress are informed by additional factors. Our findings are in agreement, extending this line of research to exploring the relationship between participant reported perceptions of ability and physiological arousal during tasks presenting a physical challenge. Similarly, across studies, the individual factors underpinning perception remain important predictors requiring further inquiry to identify.

Specific to our use case of measurement of perception of stability while performing a walking task, further research is needed to address how changes in the physiological arousal response relates to balance self-efficacy over the course of rehabilitation of following stroke. The ABC, a measure of balance self-efficacy, has been shown to be a predictor of falls [42] and community reintegration [43] post-stroke. In this study, we were unable to examine ABC as a predictor of RPS and physiological arousal due to an insufficient sample size. This highlights an opportunity for future research exploring interactions between ABC scores, RPS ratings, and physiological arousal.

Generalizability of the findings of our study are limited. Participants in the current study were in the chronic phase of recovery and had experienced most of the functional recovery expected post stroke [44], lived independently in the community, tended to have moderate level of lower extremity impairment, and reported medium to high level of self-reported balance confidence and low levels of state and trait anxiety. Therefore, future research should consider exploring these relationships in the sub-acute population when the post-stroke neural circuits are malleable and responsive to high intensity restorative treatments and perceptions of ability [45–47]. Furthermore, lesion location has shown to impact both balance and recovery of walking post stroke [48,49], and individuals with poorer walking and balance performance often report lower balance confidence, which is an independent predictor of perceived physical function, mobility, and recovery [5]. Thus, future research inclusive of a larger sample size could explore lesion location as a potential moderator of the association between participant reported perception of stability and the physiological arousal response while performing tasks that challenge standing and walking balance. This line of research has the potential to aid with interventions which aim to implement graded challenges to standing and walking balance during the active phase of stroke recovery. Furthermore, exploring these relationships specifically in people with chronic stroke living in the community with low balance confidence could provide insight into psychosocial presentations post stroke that leads to activity avoidance, sedentary behaviour and a lack of community reintegration associated with perception of walking balance [50].

## Conclusion

Our results suggest that understanding perception of balance in people with chronic stroke is complex. Participant specific characteristics are stronger predictors of perceived stability than objective measures of balance performance and physiological arousal response to balance challenge. Therefore, considering balance self-efficacy has been reported to be a predictor of activity and participation post stroke [3,51] and associated with functional outcomes following stroke [4], use of self-reported scales such as the RPS are important to measure the individual response to balance challenge post-stroke. Although our study found that repeat exposure to balance challenges did not moderate the relationship between the independent variables of task performance and physiological arousal response and the dependent variable of perceived stability, further exploration is needed. Specifically, exploring the use of these measures over the course of rehabilitation of walking balance post stroke can help us understand how these relationships evolve during recovery of independent walking.

## Author contributions

**Conceptualization:** Aishwarya Shenoy, Rebecca M. Todd, Courtney L. Pollock.

**Formal analysis:** Aishwarya Shenoy, Amy Schneeberg, Towela Tembo, Courtney L. Pollock.

**Funding acquisition:** Courtney L. Pollock.

**Investigation:** Aishwarya Shenoy, Tzu-Hsuan Peng.

**Methodology:** Aishwarya Shenoy, Towela Tembo, Tzu-Hsuan Peng.

**Project administration:** Aishwarya Shenoy, Towela Tembo.

**Supervision:** Courtney L. Pollock.

**Visualization:** Aishwarya Shenoy.

**Writing – original draft:** Aishwarya Shenoy.

**Writing – review & editing:** Rebecca M. Todd, Noah D. Silverberg, Janice J. Eng, Peyman Servati, Courtney L. Pollock.

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
