## [Decision Letter · Decision Letter 0]

10 Jul 2025

Dear Dr. Pollock,

Thank you for submitting your manuscript to PLOS ONE. After careful consideration, we feel that it has merit but does not fully meet PLOS ONE’s publication criteria as it currently stands. Therefore, we invite you to submit a revised version of the manuscript that addresses the points raised during the review process.

We look forward to receiving your revised manuscript.

Kind regards,

Yi Ding

Academic Editor

PLOS ONE

Journal Requirements:

“This work was supported in part by grants from the New Frontiers in Research Fund (Exploration grant, CLP) and the Michael Smith Foundation for Health Research (Scholar Award, CLP).”

“This work was supported in part by grants from the New Frontiers in Research Fund (Exploration grant, CLP) and the Michael Smith Foundation for Health Research (Scholar Award, CLP).”

““This work was supported in part by grants from the New Frontiers in Research Fund (Exploration grant, CLP) and the Michael Smith Foundation for Health Research (Scholar Award, CLP).””

Reviewers' comments:

Reviewer's Responses to Questions

**Comments to the Author**

1. Is the manuscript technically sound, and do the data support the conclusions?

Reviewer #1: Yes

Reviewer #2: Partly

2. Has the statistical analysis been performed appropriately and rigorously?

Reviewer #1: Yes

Reviewer #2: Yes

3. Have the authors made all data underlying the findings in their manuscript fully available?

Reviewer #1: Yes

Reviewer #2: Yes

4. Is the manuscript presented in an intelligible fashion and written in standard English?

Reviewer #1: Yes

Reviewer #2: Yes

Reviewer #1: This is an automated report for PONE-D-25-16381. This report was solicited by the PLOS One editorial team and provided by ScreenIT.

ScreenIT is an independent group of scientists developing automated tools that analyze academic papers. A set of automated tools screened your submitted manuscript and provided the report below. Each tool was created by your academic colleagues with the goal of helping authors. The tools look for factors that are important for transparency, rigor and reproducibility, and we hope that the report might help you to improve reporting in your manuscript. Within the report you will find links to more information about the items that the tools check. These links include helpful papers, websites, or videos that explain why the item is important. While our screening tools aim to improve and maintain quality standards they may, on occasion, miss nuances specific to your study type or flag something incorrectly. Each tool has limitations that are described on the ScreenIT website. The tools screen the main file for the paper; they are not able to screen supplements stored in separate files. Please note that the Academic Editor had access to these comments while making a decision on your manuscript. The Academic Editor may ask that issues flagged in this report be addressed. If you would like to provide feedback on the ScreenIT tool, please email the team at ScreenIt@bih-charite.de. If you have questions or concerns about the review process, please contact the PLOS One office at plosone@plos.org.

Reviewer #2: The authors examine, in stroke patients, how objective and subjective (RPS, related to the lab balance challenge) measures of balance relate to Electrodermal arousal (EDA), with the ultimate goal of understanding which objective measures best predict RPS. This is important since self efficacy at balance contributes to the quality of life for stroke patients. Objective and subjective measures of balance were related, while EDA was not a useful indicator (significant for RPS but small effect size). Individual effects explained even more variance in RPS, suggesting that additional factors may be critical when trying to understand and enhance the sense of self-efficacy in maintaining balance.

The authors need to do analyses including ABC, the subjective sense of balance in real life. In part since this showed a sex difference. This metric would seem, from first principle, to potentially have important implications for quality of life.

The discussion of arousal is incomplete (e.g., l.395-). EDA is only one arousal measure, with heart rate variability, cortisol, and others also capturing arousal patterns that are missed with EDA. Thus, the authors should not overstate that arousal did not relate to RPS, since only one facet of arousal was measured.

The abstract needs to be rewritten to be clearer, especially about what higher or lower in different measures indicate, and the interpretation of the relation between measures.

l.48 “repeat exposure:” does this mean first vs second test day? Is there precedent that that d1 and d2 of testing would be different?

l.57-58, 313- is unclear for “decreases in participant RPS scores relative of 2.80.” Does this mean that greater objective balance associates with reduced subjective balance?

l.63-64 what %variance explained by individual effects?

l.206 calculating baseline for EDR from 4 sec around the minimum could bias baseline measures, e.g. magnifying small dips from noise in the recording. While a reference is cited, this baseline measure requires clearer justification.

l.214 normalizing EDR to the maximum also needs good justification. Many studies use raw EDR, and normalizing could washout e.g. where people with larger EDR have less balance self-efficacy.

l.253 are Tukey post hoc corrected for multiple comparisons?

l.184-185, 300- Please discuss more the concern is that removing measures in CBM will skew the remaining findings.

Minor

l.137 change to “; and (3)”

**Do you want your identity to be public for this peer review?** For information about this choice, including consent withdrawal, please see our Privacy Policy

Reviewer #1: No

Reviewer #2: No

---

## [Author Response · Author response to Decision Letter 1]

25 Aug 2025

*Please note Response to Reviewers uploaded as a document with inclusion of formatting to assist review. All line and page numbers refer to clean version of revised manuscript.

Answers to Reviewers

Journal Requirements

Fulfilled

Funding statement has been removed from the acknowledgements section.

Updated funding statement provided below:

“This work was supported in part by grants from the New Frontiers in Research Fund (Exploration grant, CLP) and the Michael Smith Foundation for Health Research (Scholar Award, CLP). The funders had no role in study design, data collection and analysis, decision to publish, or preparation of the manuscript”

““This work was supported in part by grants from the New Frontiers in Research Fund (Exploration grant, CLP) and the Michael Smith Foundation for Health Research (Scholar Award, CLP).””

Addressed, please refer to text in comment 2 above.

Data has been made available at Borealis Dataverse Repository at the following DOI: https://doi.org/10.5683/SP3/SXVBGQ

Fulfilled; this can be found on Page 6 line 153-154, “Participants provided informed written consent before participation. This study was approved by the University of British Columbia Clinical Research Ethics Board.”

Not applicable

Reviewer #1

1. Ethics

Fulfilled

2. Inclusion/Exclusion Criteria:

Fulfilled

3. Flow Charts and Attrition

Figure created; Fig 4

4. Sex as a biological variable

Reported in Table 1

5. Subject Demographics

Reported in Table 1

6. Randomization

No randomization required; single group study

7. Blinding

Not applicable

8. Power Analysis

The following was added to the statistical Analysis section of the manuscript:

“Power for the mediation pathway was approximated by calculating power for each constituent effect (CB&M task performance→ physiological arousal; physiological arousal → RPS) using Cohen’s d and the pwr package, then combining them. Observed, standardized, and hypothetical medium-sized effects (d = 0.35) were evaluated for sensitivity.” (Page 11, Line 273-277)

9. Replication

Not required

10. Open Code

Not applicable

11. Open Data

The study data are publicly available in the Borealis Dataverse Repository at the following DOI: https://doi.org/10.5683/SP3/SXVBGQ

12. Self acknowledged Limitations

Limitations can be found in the Discussion session, Page 21, line 448; “Generalizability of the findings of our study are limited. Participants in the current study were in the chronic phase of recovery and had experienced most of the functional recovery expected post stroke [39], lived independently in the community, tended to have moderate level of lower extremity impairment, and reported medium to high level of self-reported balance confidence and low levels of state and trait anxiety. Therefore, future research should consider exploring these relationships in the sub-acute population when the post-stroke neural circuits are malleable and responsive to high intensity restorative treatments and perceptions of ability [40–42].”

13. Conflict of interest

Addressed on Page 22, line 475-476, “The authors declare no conflicts of interest.”

14. Funding Statement

Fulfilled

15. Registration Statement

Not Applicable

16. No Bar graphs of continuous data

Fulfilled

17. No rainbow color maps

Fulfilled

Reviewer #2

Comment 1: The authors need to do analyses including ABC, the subjective sense of balance in real life. In part since this showed a sex difference. This metric would seem, from first principle, to potentially have important implications for quality of life.

Response 1: The following has been added to the Discussion section of the manuscript to highlight the importance of studying this relationship in future studies.

“Specific to our use case of measurement of perception of stability while performing a walking task, further research is needed to address how changes in the physiological arousal response relates to balance self-efficacy over the course of rehabilitation of following stroke. The ABC, a measure of balance self-efficacy, has been shown to be a predictor of falls [39] and community reintegration [40] post-stroke . In this study, we were unable to examine ABC as a predictor of RPS and physiological arousal due to an insufficient sample size. This highlights an opportunity for future research exploring interactions between ABC scores, RPS ratings, and physiological arousal.” (Page 20, Lines 440-447)

Comment 2: The discussion of arousal is incomplete (e.g., l.395-). EDA is only one arousal measure, with heart rate variability, cortisol, and others also capturing arousal patterns that are missed with EDA. Thus, the authors should not overstate that arousal did not relate to RPS, since only one facet of arousal was measured.

Response 2:

In the introduction, we have described alternative methods for assessing physiological arousal and highlighted why EDA was the best measure of physiological arousal for our research question.

“Heart rate variability (HRV), cortisol production and electrodermal activity (EDA) can measure physiological arousal modulated by the autonomic nervous system (i.e., “fight or flight” response)[12,13]. EDA is a measurement of skin conductance driven exclusively by sympathetic nervous system activation [14], and is activated in conditions requiring heightened attention, state anxiety, or cognitive load [15]. EDA has shown to be a better metric to capture brief, task-related changes in sympathetic nervous system activity than metrics specific to HRV and cortisol production [16].” Page 4, Lines 101-107

We have also clarified (where applicable), that we are looking EDA as the physiological arousal measure.

“Contrary to our hypothesis physiological arousal, as measured by EDA, was not a mediator of the relationship between CB&M and RPS.” (Page 19, Line 416-417)

Comment 3: The abstract needs to be rewritten to be clearer, especially about what higher or lower in different measures indicate, and the interpretation of the relation between measures.

l.48 “repeat exposure:” does this mean first vs second test day? Is there precedent that that d1 and d2 of testing would be different?

l.57-58, 313- is unclear for “decreases in participant RPS scores relative of 2.80.” Does this mean that greater objective balance associates with reduced subjective balance?

l.63-64 what %variance explained by individual effects?

Response 3:

- Interpretation of results has been clarified.

“CB&M scores, indicating balance performance, explained 20.4% of the variability in the RPS. As CB&M task performance improved, RPS scores decreased by 2.80 (95% CI [-3.29 – -2.30]) to 3.67 (95% CI [-4.32 – -3.02]) points, indicating improved perceived stability.” (Page 2, Paragraph 3, line 57-60)

- Percentage of variance explained from individual effects has been clarified.

“Participant-specific random effects accounted for more variance in the RPS than the fixed effects of task performance and physiological arousal, explaining 46% of variance in RPS.” (Page 2, Paragraph 3, line 63-64)

Comment 4: l.206 calculating baseline for EDR from 4 sec around the minimum could bias baseline measures, e.g. magnifying small dips from noise in the recording. While a reference is cited, this baseline measure requires clearer justification.

Response 4:

We ensured that the 4 second period of the signal was taken from the most stable portion of the signal. We have clarified this by adding the following into the Methods section.

“Manual inspection was employed to ensure the epoch representative of pre-task baseline was stable and did not include large fluctuations in the signal.” (Page 8, Paragraph 2, Line 209-210)

Comment 5: l.214 normalizing EDR to the maximum also needs good justification. Many studies use raw EDR, and normalizing could washout e.g. where people with larger EDR have less balance self-efficacy.

Response 5: Normalization was performed to enable comparisons across tasks between participants. The following was added to the manuscript to justify our approach.

“While larger EDA responses reflect greater autonomic arousal within individuals, it is difficult to compare the responses between subjects as baseline levels and response amplitudes can vary between people due to non-task related factors (e.g., medications, skin properties) [33]. Therefore, for each day, the maximum task-specific physiological arousal response out of all the tasks was identified for each participant separately. This participant-specific maximum response was used to normalize response values across all tasks performed by the individual to aid interpretation. Task-specific physiological arousal response values after normalization ranged from 0%-100%, with 100% representing the task that triggered the participant’s maximum response for that day. The normalized task-specific physiological arousal response for each item in the CB&M was used for data analyses. This preserved within-person response patterns while enabling group-level modeling of the relationship between EDA and perceived stability (RPS) across tasks.” (Page 9, Line 215-226)

Comment 6: l.253 are Tukey post hoc corrected for multiple comparisons?

Response 6: The following has been added to the Statistical Analysis section of the manuscript:

“Post hoc pairwise comparisons among CBM performance levels were conducted in R using the emmeans package with Tukey’s HSD adjustment, which corrects for multiple comparisons by controlling the family-wise error rate at α = 0.05.” (Page 11, Line 263-265)

Comment 7: l.184-185, 300- Please discuss more the concern is that removing measures in CBM will skew the remaining findings.

Response 7:

We have provided additional explanation for why some of the CB&M tasks were removed from analysis.

Methods:

“During data collection, some participants either declined to attempt tasks they perceived they would be unable to perform or performed tasks in a manner that did not adequately challenge their balance (i.e., use of physical assistance), leading to a CB&M score of 0. Therefore, CB&M items were only included in analyses if 1) at least 10 participants completed the task (to ensure enough observations were present) and 2) the mean CB&M score was at least 2 (indicating that on average, participants performed the task without physical assistance).” (Page 7, Line 184-189)

Results:

“Removing these tasks from analysis ensured that the corresponding physiological arousal response to a task was specifically attributable to balance being challenged.” (Page 15, Line 323-325)

Comment 8: l.137 change to “; and (3)”

Response 8: Addressed

---

## [Decision Letter · Decision Letter 1]

4 Nov 2025

Dear Dr. Pollock,

Thank you for submitting your manuscript to PLOS ONE. After careful consideration, we feel that it has merit but does not fully meet PLOS ONE’s publication criteria as it currently stands. Therefore, we invite you to submit a revised version of the manuscript that addresses the points raised during the review process.

Consider adjusting the paper title to reflect the relatively small sample size (“preliminary”?)Discuss sex-related differences about risk factors, stroke subtypes, stroke severity, and outcomeClarify the meaning of “repeat exposure”Provide additional details / clarify linear mixed model (exposure variable / fixed effects / random intercept / grouping structure)Briefly discuss future directions (potentially including lacunar vs non-lacunar ischemic strokes)Check for additional typos, references, and minor cosmetic issues flagged in the reviewers' reports

We look forward to receiving your revised manuscript.

Kind regards,

Luca Citi, PhD

Academic Editor

PLOS ONE

Journal Requirements:

Reviewers' comments:

Reviewer's Responses to Questions

**Comments to the Author**

Reviewer #2: (No Response)

Reviewer #3: All comments have been addressed

Reviewer #4: All comments have been addressed

Reviewer #5: All comments have been addressed

2. Is the manuscript technically sound, and do the data support the conclusions?

Reviewer #2: Yes

Reviewer #3: Yes

Reviewer #4: Yes

Reviewer #5: Yes

3. Has the statistical analysis been performed appropriately and rigorously?

Reviewer #2: Yes

Reviewer #3: Yes

Reviewer #4: Yes

Reviewer #5: I Don't Know

4. Have the authors made all data underlying the findings in their manuscript fully available?

Reviewer #2: Yes

Reviewer #3: Yes

Reviewer #4: Yes

Reviewer #5: Yes

5. Is the manuscript presented in an intelligible fashion and written in standard English?

Reviewer #2: Yes

Reviewer #3: Yes

Reviewer #4: (No Response)

Reviewer #5: Yes

Reviewer #2: now good. The paper provides new data, which are somewhat preliminary but help advance use of biometrics as indicators for risks like falls during stroke

Reviewer #3: The authors present the results of a potentially interesting revised version of an observational clinical study showing that ability to perform a walking balance task (CB&M task performance score) and the underlying physiological arousal response are independent predictors of perception of balance in people with chronic stroke as measured by the Rate of Perceived Stability. The study can improved if the following minor considerations are addressed:

1.Due to the small size of the study (n=27 patients) the title should clearly mention “preliminary findings”.

2.It would be useful to clearly mention in the Introduction section that women differ from men in the distribution of risk factors and stroke subtypes, stroke severity, and outcome (see data and comment on the study published in Cerebrovasc Dis 2025;54(1):11-19. doi: 10.1159/000536436. Epub 2024 Jan 29. PMID: 38286114).

3.It would be interesting to know the different stroke subtypes (cardioembolic stroke, lacunar infarct, infarct of unusual etiology, essential cerebral infarct, atherothrombotic infarct, intracerebral hemorrhage) in the study population.

4.It would be interesting to add in the text that an essential line of future research would be precisely to evaluate the impact of the differences on this topic between lacunar and non-lacunar ischemic strokes. This recommendation is because the pathophysiology, prognosis, and clinical features of ischemic lacunar strokes are different from other acute ischemic cerebrovascular diseases (see and include this supporting reference: Neuroepidemiology 2010;35:231-236),

5.A brief final comment on other possible lines of future research on the presented topic would be appreciated.

6.Please check references #26, #34 and #46

Reviewer #4: Line 97 to 99 there is no figure but the figure caption is there

Was participant subjected to walking 10 meters walk this before recruitment of it was base on self report?

Line 249-252Move to the section where you explained the CB&M before this paragraph

Linear Mixed Models line 253-261 is unclear, and I am having difficulty following exactly what was done. Please clearly specify the exposure variable(s) and the fixed effects included in your model. In addition, clarify how the random intercept was formulated and what grouping structure it represents.

"response). Power for

274 the mediation pathway was approximated by calculating power for each constituent effect (CB&M

275 task performance→ physiological arousal; physiological arousal → RPS) using Cohen’s d and the

276 pwr package, then combining them. Observed, standardized, and hypothetical medium-sized

277 effects (d = 0.35) were evaluated for sensitivity." -The approach described here approximates power by calculating it separately for each constituent effect and then combining them. This should be noted as a limitation, because the true power for the mediation pathway is typically lower, both paths must be significant for the indirect effect to hold. As a result, this approach may overestimate the true mediation power.

Reviewer #5: Thank you for your responses. All the issues requested by previous reviewers have been corrected. I only found one question that I couldn't find an answer to.

Comment 3:

l.48 “repeat exposure:” does this mean first vs second test day? Is there precedent that that d1 and d2 of testing would be different?

**Do you want your identity to be public for this peer review?** For information about this choice, including consent withdrawal, please see our Privacy Policy

Reviewer #2: No

Reviewer #3: **Yes: ** Dr. Adrià Arboix

Reviewer #4: No

Reviewer #5: No

---

## [Author Response · Author response to Decision Letter 2]

27 Nov 2025

Dear Dr. Luca Citi,

We thank you and the reviewers for your thoughtful and constructive feedback of our manuscript entitled “Relationship between the Rate of Perceived Stability, electrodermal activity and task performance during balance challenges in chronic stroke”. We have carefully considered all comments and revised the manuscript accordingly.

Editor Comments:

Comment #1: Consider adjusting the paper title to reflect the relatively small sample size (“preliminary”?).

Response 1: We believe that the term “preliminary findings” may not accurately reflect the nature of this work. This manuscript presents the complete analysis from the full sample of 27 participants, rather than an early-phase or partial dataset. The findings are intended to inform future research rather than represent interim results. As suggested by the reviewers, we have noted that a larger sample size would enable the exploration of additional relationships among the variables examined in this study.

“Thus, future research inclusive of a larger sample size could explore lesion location as a potential moderator of the association between participant reported perception of stability and the physiological arousal response while performing tasks that challenge standing and walking balance.” Page # 22, Line 478-481

Comment #2: Discuss sex-related differences about risk factors, stroke subtypes, stroke severity, and outcome.

Response 2: We have added additional information regarding sex related differences.

“Sex differences have been previously reported in stroke subtypes, severity and recovery outcomes [21,22]. Although males have higher incidence of first-ever stroke compared to females [23], prognosis and recovery outcomes related to walking are reported to be poorer in females than males despite adjustment for baseline differences in age, pre-stroke function and comorbidities [22,24,25].” Page #5, Line # 128-132

Comment #3: Clarify the meaning of “repeat exposure”

Response #3: We have clarified the meaning of repeat exposure in both the abstract and the introduction.

Definition of repeated exposure in Abstract:

“This study aims to examine the relationships between EDA, task performance ability, and RPS in individuals with chronic stroke performing walking balance challenges. Further, the study explores how sex and repeat exposure (repeat performance of task) moderate these relationships.” Page #2, Line 43-46

Definition of repeated exposure in Introduction:

“This aim is addressed through three sequential objectives: 1) investigate the relationship between walking balance task performance scores and RPS ratings, and determine whether sex or repeated exposure (repeat performance of tasks) across days moderates this relationship” Page #5, Line #138-141

Comment #4: Provide additional details / clarify linear mixed model (exposure variable / fixed effects / random intercept / grouping structure)

Response #4: We have clarified the linear mixed model.

“Two different models were used for each objective, where the predictor (independent variable) was CB&M Task Performance for Objective 1 and physiological arousal for Objective 2. For both models (Model 1. CB&M task performance and Model 2. Physiological arousal for each CB&M task), each CB&M task was treated as a separate observation. Because multiple tasks were completed by each participant, the data had a hierarchical structure with tasks nested within participants. This grouping structure was modeled by including a random intercept for participant to account for within-subject correlation. Sex and repeated exposure (Day 1 vs Day 2) were explored as potential effect moderators by including them as separate interaction terms in both models, nested models were compared via likelihood ratio tests (p > 0.05 considered significant). Tukey post hoc test was performed to assess pairwise differences between groups for all significant interaction terms.” Page #11, Line #268-278.

Comment #5: Briefly discuss future directions (potentially including lacunar vs non-lacunar ischemic strokes). Thoughts: discuss impact of type of lesion on perception of balance as a future area for study/exploration

Response #5: The following has been added to future lines of research;

“Furthermore, lesion location has shown to impact both balance and recovery of walking post stroke [48,49], and individuals with poorer walking and balance performance often report lower balance confidence, which is an independent predictor of perceived physical function, mobility, and recovery [5] .Thus future research inclusive of a larger sample size could explore lesion location as a potential moderator of the association between participant reported perception of stability and the physiological arousal response while performing tasks that challenge standing and walking balance.” Page #22, Line 475-481.

Comment #6: Check for additional typos, references, and minor cosmetic issues flagged in the reviewers' reports

Response #6: Manuscript has been revised for typos and cosmetic issues.

REVIEWER #2:

Now good. The paper provides new data, which are somewhat preliminary but help advance use of biometrics as indicators for risks like falls during stroke

Response to Reviewer #2: All comments addressed, no response required.

REVIEWER #3

The authors present the results of a potentially interesting revised version of an observational clinical study showing that ability to perform a walking balance task (CB&M task performance score) and the underlying physiological arousal response are independent predictors of perception of balance in people with chronic stroke as measured by the Rate of Perceived Stability. The study can improved if the following minor considerations are addressed:

Comment #1: Due to the small size of the study (n=27 patients) the title should clearly mention “preliminary findings”.

Response #1: Thank you for your comment and suggestion to clarify the scope of the study in the title. We believe that the term “preliminary findings” may not accurately reflect the nature of this work. This manuscript presents the complete analysis from the full sample of 27 participants, rather than an early-phase or partial dataset. The findings are intended to inform future research rather than represent interim results. As suggested by the reviewers, we have noted that a larger sample size would enable the exploration of additional relationships among the variables examined in this study.

“Furthermore, lesion location has shown to impact both balance and recovery of walking post stroke [48,49], and individuals with poorer walking and balance performance often report lower balance confidence, which is an independent predictor of perceived physical function, mobility, and recovery [5] .Thus, future research inclusive of a larger sample size could explore lesion location as a potential moderator of the association between participant reported perception of stability and the physiological arousal response while performing tasks that challenge standing and walking balance.” Page #22, Line 475-481.

Comment #2: It would be useful to clearly mention in the Introduction section that women differ from men in the distribution of risk factors and stroke subtypes, stroke severity, and outcome (see data and comment on the study published in Cerebrovasc Dis 2025;54(1):11-19. doi: 10.1159/000536436. Epub 2024 Jan 29. PMID: 38286114).

Response #2:

“Sex differences have been previously reported in stroke subtypes, severity and recovery outcomes [21,22]. Although males have higher incidence of first-ever stroke compared to females [23], prognosis and recovery outcomes related to walking are reported to be poorer in females than males despite adjustment for baseline differences in age, pre-stroke function and comorbidities [22,24,25].” Page #5, Line # 128-132

Comment #3:

It would be interesting to know the different stroke subtypes (cardioembolic stroke, lacunar infarct, infarct of unusual etiology, essential cerebral infarct, atherothrombotic infarct, intracerebral hemorrhage) in the study population.

Response #3:

“Inclusion criteria included: at least one year-since stroke (occurrence of stroke confirmed by medical records), hemiparesis post-stroke, the ability to walk at least 10 metres with or without a walking aid (confirmed during initial assessment)”. Page #7, Line #157-160

Comment #4: It would be interesting to add in the text that an essential line of future research would be precisely to evaluate the impact of the differences on this topic between lacunar and non-lacunar ischemic strokes. This recommendation is because the pathophysiology, prognosis, and clinical features of ischemic lacunar strokes are different from other acute ischemic cerebrovascular diseases (see and include this supporting reference: Neuroepidemiology 2010;35:231-236),

Response #4: The following has been added to the limitation section of the Discussion, and recommended references have been added.

“Furthermore, lesion location has shown to impact both balance and recovery of walking post stroke [48,49], and individuals with poorer walking and balance performance often report lower balance confidence, which is an independent predictor of perceived physical function, mobility, and recovery [5] .Thus, future research inclusive of a larger sample size could explore lesion location as a potential moderator of the association between participant reported perception of stability and the physiological arousal response while performing tasks that challenge standing and walking balance.” Page #22, Line 475-481.

Comment #5: A brief final comment on other possible lines of future research on the presented topic would be appreciated.

Response #5: This has been addressed partly in the comment above along with some changes in wording in the Discussion.

“Therefore, future research should consider exploring these relationships in the sub-acute population when the post-stroke neural circuits are malleable and responsive to high intensity restorative treatments and perceptions of ability [45–47]. Furthermore, lesion location has shown to impact both balance and recovery of walking post stroke [48,49], thus future research inclusive of a larger sample size could explore lesion location as a potential moderator of the association between participant reported perception of stability and the physiological arousal response while performing tasks that challenge standing and walking balance. This line of research has the potential to aid with interventions which aim to implement graded challenges to standing and walking balance during the active phase of stroke recovery. Furthermore, exploring these relationships specifically in people with chronic stroke living in the community with low balance confidence could provide insight into psychosocial presentations post stroke that leads to activity avoidance, sedentary behaviour and a lack of community reintegration associated with perception of walking balance [50].” Page #22, Lines 472-481

Comment #6: Please check references #26, #34 and #46

Response #6: Formatting for references have been fixed.

Reference #26 is now #29, Reference #34 is now #37, Reference #46 is now #51

REVIEWER #4:

Comment #1: Line 97 to 99 there is no figure but the figure caption is there

Response #1: Journal specific submission requirements state to not include the figure within the manuscript but to upload them as an attachment during the submission process.

Comment #2: Was participant subjected to walking 10 meters walk this before recruitment of it was base on self report?

Response #2:

“Inclusion criteria included: at least one year-since stroke (confirmed by medical records), hemiparesis post-stroke, the ability to walk at least 10 metres with or without a walking aid (confirmed during initial assessment).” Page #7, Line 157-160

Comment #3: Line 249-252 Move to the section where you explained the CB&M before this paragraph

Response #3: Has been moved to section where we talk about the CB&M. Page #8 Line 197-205

Comment #4: Linear Mixed Models line 253-261 is unclear, and I am having difficulty following exactly what was done. Please clearly specify the exposure variable(s) and the fixed effects included in your model. In addition, clarify how the random intercept was formulated and what grouping structure it represents.

Response #4: Section has been reworded to clarify the variables and effects.

“To address Objectives 1 and 2, Linear Mixed Models (LMMs) using the lmer function from the lme4 package [35]. Two different models were used for each objective, where the predictor (independent variable) was CB&M Task Performance for Objective 1 and physiological arousal for Objective 2. For both models (Model 1. CB&M task performance and Model 2. Physiological arousal for each CB&M task), each CB&M task was treated as a separate observation. Because multiple tasks were completed by each participant, the data had a hierarchical structure with tasks nested within participants. This grouping structure was modeled by including a random intercept for participant to account for within-subject correlation. Sex and repeated exposure (Day 1 vs Day 2) were explored as potential effect moderators by including them as separate interaction terms in both models, nested models were compared via likelihood ratio tests (p > 0.05 considered significant). Tukey post hoc test was performed to assess pairwise differences between groups for all significant interaction terms.” Page #11, Line #267-278.

Comment #5: “Power for the mediation pathway was approximated by calculating power for each constituent effect (CB&M task performance→ physiological arousal; physiological arousal → RPS) using Cohen’s d and the pwr package, then combining them. Observed, standardized, and hypothetical medium-sized effects (d = 0.35) were evaluated for sensitivity." -The approach described here approximates power by calculating it separately for each constituent effect and then combining them. This should be noted as a limitation, because the true power for the mediation pathway is typically lower, both paths must be significant for the indirect effect to hold. As a result, this approach may overestimate the true mediation power.

Response #5: Further explanation is provided;

“However, power for the mediation analysis was approximated by calculating it separately for each constituent path and then combining them. This approach, while practical, may overestimate the true power for detecting the mediation effect. Consequently, the mediation analysis should be considered exploratory.” Page #21, Line 443-447

REVIEWER #5:

Comment #1: l.48 “repeat exposure:” does this mean first vs second test day? Is there precedent that that d1 and d2 of testing would be different?

Response #1: The definition of repeat exposure and precedent has been clarified in both the abstract and the introduction.

Abstract: “Measurement of electrodermal activation (EDA), modulated by the autonomic nervous system, during perturbations to standing balance reflects the physiological arousal (‘fight or flight’) response of the individual as they maintain or recover their balance. Repeat performance of a balance task has been shown to result in habituation of EDA within a single session [18]; however, studies have yet to test whether similar habituation occurs when the same balance tasks are repeated across different days.” Page 2, Line 38-43

Introduction: “Furthermore, as clinical walking balance interventions extend over multiple sessions, it would be important to determine whether these relationships change with repeated exposure to the walking challenges on subsequent days to enhance the clinical utility and interpretability of treatment intensity measures. Repeat performance of a balance task has been shown to result in habituation of EDA within a single session [18]; however, studies have yet to test whether similar habituation occurs when the same balance tasks are repeated across different days.” Page #5, Line #121-127

---

## [Editor Report · Decision Letter 2]

2 Dec 2025

Relationship between the rate of perceived stability, electrodermal activity and task performance during balance challenges in chronic stroke

PONE-D-25-16381R2

Dear Dr. Pollock,

We’re pleased to inform you that your manuscript has been judged scientifically suitable for publication and will be formally accepted for publication once it meets all outstanding technical requirements.

Kind regards,

Luca Citi, PhD

Academic Editor

PLOS ONE

---

## [Editor Report · Acceptance letter]

PONE-D-25-16381R2

PLOS One

Dear Dr. Pollock,

I'm pleased to inform you that your manuscript has been deemed suitable for publication in PLOS One. Congratulations! Your manuscript is now being handed over to our production team.

Kind regards,

on behalf of

Dr. Luca Citi

Academic Editor

PLOS One